# The Effect of Co-Fermentation with *Lactobacillus plantarum* HLJ29L2 and Yeast on Wheat Protein Characteristics in Sourdough and Crackers

**DOI:** 10.3390/foods12030555

**Published:** 2023-01-27

**Authors:** Liping Hu, Yue Li, Xiang Huang, Chaodong Du, Dejian Huang, Xiumei Tao

**Affiliations:** 1State Key Laboratory of Food Science and Technology, Jiangnan University, Wuxi 214122, China; 2School of Food Science and Technology, Jiangnan University, Wuxi 214122, China; 3International Joint Laboratory on Food Safety, Jiangnan University, Wuxi 214122, China; 4Department of Food Science and Technology, National University of Singapore, Singapore 117542, Singapore

**Keywords:** co-fermentation, sourdough, protein, fermented cracker, protein digestibility

## Abstract

Sourdough fermentation has been widely used in food products. However, there has been limited study of the effect of co-fermentation with lactic acid bacteria and yeast on the dough and cracker products. In this study, the influence of co-fermentation with *Lactobacillus plantarum* HLJ29L2 (LP HLJ29L2) and yeast on wheat protein digestibility of cracker was studied, and the mechanism of the protein changes in sourdough during fermentation was further explored. Co-fermentation with LP HLJ29L2 and yeast (DN-1) strongly improved the protein digestibility of cracker. At the same time, the content of free amino acids in DN-1 crackers increased by 20%. Co-fermentation also had significant effect on the sourdough during fermentation. The SDS-soluble protein content in sourdough was increased, and large molecule proteins were significantly reduced in the DN-1 sourdough. This was due to the fact, that LP HLJ29L2 grew rapidly during co-fermentation and produced more organic acids, which led to an increase in protease activity in sourdough and promoted the degradation of protein by proteases. The results of this study provide an important theoretical basis for the application of lactic acid bacteria and yeast co-fermentation in crackers.

## 1. Introduction

Lactic acid bacteria (LAB) are regarded as beneficial microorganisms. Fermentation with LAB has a long history, and it still plays an important role today [1]. Wheat is the most widely planted crop in the world, and it is the most important food crop in many countries. In recent years, fermentation by yeast combined with LAB has been used in wheat dough to improve the sensory qualities and nutritional value of baked products [2,3,4]. The microbiota of sourdough is composed of LAB and yeasts. LAB are the most numerous and metabolically active organisms in sourdough [5]. The benefits of sourdough fermentation are primarily related to LAB metabolic activities such as proteolysis, flavor compound synthesis, and the unique symbiotic relationship of certain hetero- and homo-fermentative LAB with certain yeasts [6,7]. It was proposed that sourdough fermentation resulted in the solubilization and depolymerization of gluten macropolymers [5,8,9].

During fermentation, the metabolism of yeast produces a large amount of gas, while LAB mainly produce lactic, malic, and acetic acids [10]. This would cause a drop in the pH of sourdough, which could activate protease activity and promote the degradation of proteins into peptides and amino acids [11]. The main proteolytic activity was first attributed to endogenous flour enzymes, such as aminopeptidase and endopeptidase [12]. Additionally, enzymes in LAB and other microorganisms also play an important role [5]. As one of the macromolecular substances, protein is an integral part of our daily diet. Ketnawa et al. [13] studied the protein digestibility of soaked, boiled, and fermented soybeans. The results showed that soybeans after the fermentation step had a higher protein digestibility, amino acids content, and potential bioactivity compared to those after the boiling and soaking steps [13]. These studies indicated that fermentation with lactic acid bacteria could cause the degradation of protein. However, due to the difference in recipes and processing of products, the digestibility of protein was significantly different between different studies [14]. The mechanism of the protein changes during fermentation needs to be further explored. Moreover, proteolysis during sourdough fermentation with LAB for crackers has been poorly investigated, and the mechanism of improving biscuit quality using acid dough fermentation is lacking an in-depth exploration, which greatly limits the application of acid dough fermentation technology in biscuit industry and the improvement of biscuit quality. At present, the single yeast fermentation method can hardly meet the demand of consumers for qualitative cookies, and exploring the application of lactic acid bacteria and yeast co-fermentation in cookie products will become a new trend in market research.

In this study, crackers were prepared from dough fermented with fresh yeast and a kind of *Lactobacillus plantarum* (LP HLJ29L2). The protein digestibility and free amino acids content of the crackers were evaluated. In order to further study the mechanism of the protein changes in sourdough during fermentation, changes in molecular weight of protein content and protease activity, the variation in pH value and organic acids, and the growth of LP HLJ29L2 and yeast in sourdoughs during fermentation were explored. Our study will provide a theoretical basis for the application of lactic acid bacteria and yeast mixed fermentation in crackers.

## 2. Materials and Methods

### 2.1. Materials

The LAB strain (*Lactobacillus plantarum* HLJ29L2) was isolated from pickle in Zhaodong City, Heilongjiang, North of China. Industrial compressed yeast (*Saccharomyces cerevisiae*) was obtained from Angel Yeast Co. (Yichang, Hubei, China). The flour utilized for producing sourdoughs and crackers was a commercial-type wheat flour (moisture, 13.1%; protein, 9.7%; ash, 0.6% (of dry basis); fat, 1.1%), and the other cracker ingredients used were fresh yeast, whole wheat flour, milk powder, shortening, leavening agent, and salt, which were all obtained from Mondelēz Shanghai Food Corporate Management Co. Ltd.

### 2.2. Formula and Preparation of Fermented Crackers

Ingredients of the fermented crackers are listed in the Appendix A (Table A1). A mixer was used to combine the sourdough ingredients, and primary fermentation was carried out in a fermentation box at 35 °C and 80% relative humidity for 19 h. Then sourdough with the remaining ingredients continued fermentation at 30 °C and 80% relative humidity for 4 h. The samples were rolled repeatedly to 1 mm and molded with a 5 cm × 5 cm molder. The patches were baked in an oven at 220 °C for 6 min. Fermented crackers with only yeast followed the same steps as LAB crackers, except for the addition of LP HLJ29L2. The fermented dough and biscuits were lyophilized (48 h) and milled to pass through a 30 mesh sieve for further analysis.

In this study, the dough or crackers without yeast and LP HLJ29L2 co-fermentation were named N-L-Y dough or crackers; the dough or crackers with yeast fermentation alone were named yeast dough or crackers and the sourdough or crackers with LP HLJ29L2 and yeast co-fermentation was named DN-1 sourdough or cracker, respectively.

### 2.3. Protein Digestibility of Fermented Crackers

The simulated digestion methods described were used with modifications to estimate the digestibility in the stomach and intestine [15,16,17]. Powdered cracker (1.00 g) was dissolved in 4 mL of simulated gastric fluid (SGF), containing 3 mL of porcine pepsin stock solution (7 mg/mL). SGF was prepared from SGF electrolyte stock solution (containing pepsin from porcine gastric mucosa 250 U/mg, Sigma), by adding 2.5 μL of 0.3 mol/L CaCl_2_ and 0.2 mL of 1 mol/L HCl to reach pH 3.0, and water to reach a total volume of 10 mL. The mixture was stirred for 2 h to ensure complete dissolution. Then 10 mL of gastric chyme was mixed with 8 mL of SIF electrolyte stock solution and 2.0 mL of a pancreatin solution (800 U/mL), which was prepared from SIF electrolyte stock solution based on trypsin activity (pancreatin from porcine pancreas, Sigma), by adding 20 μL of 0.3 mol/L CaCl_2_ and 0.15 mL of 1 mol/L NaOH to reach pH 7.0, and water to reach a total volume of 20 mL.

To study the digestion behavior, the reaction of the stomach and intestine was stopped by placing the samples in a boiling water bath for 10 min. Then the mixture was centrifuged at 5000 rpm for 20 min with an equal volume of 30% TCA [18]. The supernatant was discarded and the protein content of the precipitation was determined using Kjeldahl nitrogen [19]. The percentage of digested samples was calculated using the following equation.
(1)$$\mathrm{Protein}\;\mathrm{digestibility}\;(\%)=\frac{\left(P_{T}-P_{C}\right)}{P_{T}}\times 100$$

*P_T_* denotes the total protein content of fermented crackers. *P_C_* denotes protein content of the precipitate in the digestion stage.

### 2.4. Free Amino Acid Analysis of Fermented Crackers

Free amino acid analysis of cracker samples was performed using an automatic amino acid analyzer according to a modified method [11]. Samples were mixed with 5% TCA and filled to 50 mL with water. After ultrasonic extraction for 20 min at ambient temperature and filtering the solution for 2 h, the liquid (1 mL) was centrifuged at 15,000 rpm for 30 min. The supernatant (400 μL) was collected in a liquid sample bottle.

Free amino acid analysis was performed with an Agilent 1100 series HPLC system (Agilent Technologies, Santa Clara, CA, USA). The sample was separated on a Hyoersil ODS column (250 mm × 4.6 mm, 5 µm, Agilent) at 40 °C. The mobile phase consisted of 40% acetonitrile, 40% methanol, and 20% sodium acetate at a flow rate of 1.0 mL/min. The ultraviolet detection wavelengths used were 338 nm and 262 nm. The injection volume was 20 μL.

### 2.5. Analysis of Protein in Sourdoughs Extracts during Fermentation

Sourdough powder (1.00 g) was placed in a sample bottle with 10 mL of PBS (0.05 mol/L, pH 6.9, 2.0% SDS). Mixing and stirring were performed at room temperature for 3 h until SDS-extractable proteins were fully dissolved. Then centrifugation at 5000 rpm for 10 min and separation the pellet and supernatant followed.

The supernatant: A 0.45 μm needle organic filter membrane was used to strain the filtrate. The filtrate was collected in a liquid sample bottle using SE–HPLC for analysis [20].

The precipitate: The content of SDS-insoluble proteins was determined using GB 5009.5-2016, which was considered a glutenin macropolymer (GMP).

### 2.6. Determination of Protease Activity in Sourdoughs

The protease activity in sourdough was determined according to Thiele [21]. First, 5 g sample was placed in 10 mL of 7% perchloric acid and stored at 4 °C for 12 h. After that, the sample was centrifuged at 8000 rpm for 10 min, and the supernatant was stored. The supernatant (200 μL) was neutralized with 1.5 vol of 3 mol/L KCl to precipitate perchloric acid, stored for 1 h at 4 °C, and centrifuged at 8000 rpm for 10 min again. To a 200 μL sample 100 uL of reagent X (50 g of Na_2_HPO_4_·2H_2_O, 60 g of KH_2_PO_4_, 0.5 g of ninhydrin, 3 g of fructose, 1 L of H_2_O, pH 6.7) and 190 μL of H_2_O was added. The mixture was heated for 16 min in boiling water, and cooled down quickly to room temperature with cooling water. Following that, 500 μL of reagent Y (600 mL of H_2_O, 400 mL of 96% ethanol, 2 g of KIO_3_) was added before measuring the absorbance of the sample at 570 nm. A calibration curve was prepared for each determination using glycine.

### 2.7. Determination of pH and Organic Acids in Sourdoughs

The pH and TTA (total titratable acidity) were measured during sourdough fermentation as previously described [22]. First, 10 g of sample was placed in a beaker, homogenized with 90 mL of sterile distilled water, and stirred for approximately 30 min. The pH was recorded and then the solution was titrated with 0.1 mol/L NaOH to pH 8.1. The TTA represents the volume of 0.1 mol/L NaOH consumed to titrate 10 g of the fresh dough to pH 8.1. All tests were repeated at least three times.

For the measurement of organic acids, the samples were weighed to 1 g, mixed with 0.05% of H_3_PO_4_ to 15 mL, and centrifuged at ambient temperature and 10,000 rpm for 7 min. The supernatant was passed through a 0.22 μm needle organic filter membrane. Tests were repeated at least three times.

### 2.8. Growth of LP HLJ29L2 and Yeast during Sourdough Fermentation

During four sourdough fermentations, LP HLJ29L2 and yeast concentrations were measured. To achieve this, 10 g of sourdough was put into a sterile beaker, and homogenized with 90 mL of sterile normal saline. Appropriate dilution gradients were chosen after diluting the solution in a 10-fold gradient. The dilution solution was placed on MRS agar with cycloheximide to evaluate the growth of LP HLJ29L2. The plates were placed at 37 °C for 48 h and recorded. Then the same dilution solution was placed on YPD agar with chloramphenicol to evaluate the growth of yeast. The plates were placed at 28 °C for 48 h and observed [2].

### 2.9. Statistical Analysis

Every experiment was performed in triplicate, and the results are expressed as mean ± standard deviation. All statistical analyses were carried out with SPSS 23.0 (IBM, New York, NY, USA), and differences were considered significant at *p* < 0.05.

## 3. Results and Discussion

### 3.1. Protein Digestibility of Fermented Crackers

The proteins in fermented crackers are hydrolyzed into peptides of various sizes during digestion in the stomach due to acid hydrolysis and pepsin proteolysis. During intestinal digestion, the proteins are further degraded into small molecules of peptides and amino acids, which are easily absorbed and utilized by the body.

Protein digestibility of all three fermented crackers increased as the digestion stage progressed (Figure 1). Before digestion (0 min), there was a slight difference in digestibility between N-L-Y crackers, yeast crackers and DN-1 crackers with values of 2.6%, 3.0%, and 3.8%, respectively. The main reason may be related to the metabolic activities of LP HLJ29L2 and yeast during fermentation. The higher protein digestibility in DN-1 crackers could be owed to the large number of peptides and amino acids generated by the fermentation of LP HLJ29L2. Meanwhile, the slight difference in protein digestibility between N-L-Y and yeast crackers might be related to the growth and metabolism of yeast in sourdough, as yeast would uptake and release amino acids of dough. The protein digestibility of DN-1 crackers in the stomach phase (120 min) was around 17%, while the protein digestibility of N-L-Y and yeast crackers was around 9% and 15%, respectively. After digestion (120 min), the protein digestibility of DN-1 crackers was around 86% in the intestine. DN-1 crackers had the highest protein digestibility, followed by yeast and N-L-Y crackers. This result indicated that, compared to N-L-Y crackers, crackers prepared with yeast fermentation could help to improve protein digestibility, while crackers with LP HLJ29L2 and yeast co-fermentation were more easily digested by pepsin and trypsin, which contributed to significantly increasing the protein digestibility. Similar results have been reported by Ketnawa et al. [13]. A significant increase in the protein digestibility of fermented soybean was reported compared to soaked and boiled soybeans, owing to the large amounts of soluble peptides produced by LP HLJ29L2 fermentation. By hydrolyzing proteins into more soluble and small products, the fermentation can positively impact protein digestibility [23].

### 3.2. Analysis of Free Amino Acids in Fermented Crackers

Free amino acid contents were determined using HPLC. The results showed that crackers prepared with co-fermentation of LP HLJ29L2 and yeast had a higher content of free amino acids than N-L-Y and yeast crackers. The total free amino acid content of N-L-Y, yeast, and DN-1 crackers was 147 mg/100 g, 162 mg/100 g, and 209 mg/100 g, respectively. The total content of essential amino acids in N-L-Y and yeast crackers was 51 mg/100 g and 43 mg/100 g, accordingly. In DN-1 crackers, total free amino acid levels were much higher than in N-L-Y and yeast crackers, and an increase in the total essential amino acid content was observed at the same time. Fermentation-induced proteolysis was probably linked to the activation of endogenous proteases in acidic conditions [24].

In contrast to N-L-Y and yeast crackers, DN-1 crackers had a higher content of Arg, Val, Phe, Leu, and Pro, but the contents of Cys, Met, Ile, and Lys decreased significantly. The increased contents are due to the hydrolysis of protein by proteolytic enzymes. While the decreased content is due to microorganisms that require Cys, Met, Ile, and Lys amino acids for growth [4]. The results suggested that LP HLJ29L2 and yeast played a key role in the absorption and release of amino acids during fermentation. It was reported that the variations in each individual FAA were different in the fermentation process [11]. Furthermore, compared with N-L-Y crackers, the content of Gaba in yeast crackers increased slightly, while the content of Gaba in DN-1 crackers significantly increased.

### 3.3. Determination of SDS-Insoluble and SDS-Soluble Proteins in Sourdoughs

From the above results, the increase in protein digestibility and free amino acids in DN-1 crackers may be related to protein degradation in sourdough. During dough fermentation, the growth and metabolism of microorganisms are associated with protease activity. Therefore, the changes in molecular weight of protein content and protease activity, the growth of LP HLJ29L2 and yeast in sourdoughs during fermentation, as well as the changes in pH value and organic acids were further studied.

The content of SDS-insoluble proteins is shown in Table 1. After 19 h of fermentation, the content of GMP in N-L-Y, yeast and DN-1 sourdoughs was reduced to 0.55 g/100 g, 0.58 g/100 g, and 0.51 g/100 g, respectively. However, from the results of the depolymerization degree, it could be inferred that the degradation of GMP by microorganism fermentation was not obvious. The SDS-soluble protein content was further analyzed.

Molecular weight distributions of SDS-soluble proteins are depicted in Figure 2. The chromatograms were divided into three regions: P1 (large glutenin polymers with Mw larger than 91 kDa), P2 (large monomers with Mw from 10 kDa to 91 kDa), and P3 (peptides and amino acids with Mw below 10 kDa). The areas of P1, P2, and P3 of the chromatogram were calculated for the samples [25,26]. As shown in Figure 2a,b, before fermentation (0 h) the proportion of the molecular weight of three kinds of sourdoughs was not significantly different. After fermentation for 19 h, the area ratios of P1 decreased and P3 increased in all the sourdough samples (Figure 2c,d). The DN-1 sourdough had the lowest P1 (%) and the highest P3 (%). This showed that compared to the other two samples (N-L-Y and yeast), LP HLJ29L2 might degrade the proteins in the sourdough. The observation was in agreement with the results of protein digestibility in the crackers (0 h in stomach) (Figure 1).

### 3.4. Determination of Protease Activity of Fermented Sourdoughs

During the fermentation process, the acid production of LP HLJ29L2 is relatively strong. The proteases in the flour and LP HLJ29L2 are activated in the low pH environment, and then the protease can degrade the proteins into peptides and amino acids, which increases the content of α-amino acid nitrogen. The α-amino nitrogen concentrations at different fermentation times of the sourdoughs are shown in Figure 3a. The α-amino nitrogen concentrations have increased to varying degrees during fermentation. However, the α-amino nitrogen content of the N-L-Y dough did not increase significantly. In the dough fermented with yeast, the content of α-amino nitrogen increased slowly, while the α-amino nitrogen concentration increases significantly in the sourdough co-fermented with LP HLJ29L2 and yeast (DN-1 sourdough). The final α-amino nitrogen concentration was the lowest in the N-L-Y dough and the highest in DN-1 sourdough. The pH of N-L-Y dough and yeast dough was probably not low enough to reach the optimum pH for protein enzymes. The results were consistent with the results of free amino acids from crackers (Table 2). The change in protease activity was consistent with the results of the determination of SDS-insoluble and SDS-soluble proteins. Proteolysis caused by fermentation might be related to the activation of endogenous proteases under acidic conditions, which would significantly improve the protein digestibility and the content of free amino acids in crackers [27,28].

### 3.5. pH and TTA Measurement in Sourdoughs

The growth of LP HLJ29L2 and yeast is closely related to the change in pH in sourdoughs. pH and TTA levels of the sourdough fermented by N-L-Y, yeast and LP HLJ29L2 are shown in Figure 3b. During fermentation, the pH of DN-1 sourdoughs was higher than that of N-L-Y and yeast, and the values of TTA from LP HLJ29L2 were much higher than from the others. The pH of N-L-Y and yeast sourdough dropped slowly during fermentation. Acidification of DN-1 sourdough was much faster than that of N-L-Y and yeast sourdough. In the first 3 h of fermentation, the pH of the DN-1 sourdough dropped slowly, and the pH dropped rapidly during the next 3–12 h. Finally, the pH was relatively stable at around 12 h of fermentation. The pH value of the DN-1 sourdough changed from 6.12 to 4.04, whereas TTA developed to 10.30. This was due to the large amount of organic acids produced by LAB in DN-1 during fermentation.

The pH of sourdoughs incubated with LP HLJ29L2 and yeast were similar to the values reported previously [22,29]. In those studies, the pH value of fermented dough co-cultured with *S. cerevisiae* and *L. plantarum* stopped decreasing after 12 h of fermentation, and the changes in pH and TTA were not significant after 12 h of fermentation. This is because the lactic acid bacteria entered the stable growth stage after 12 h, and the acid production did not increase and remained relatively stable. In the early stages of fermentation, LP HLJ29L2 may need to adapt to the environment of the sourdough, and the pH changes slowly. As the fermentation time increased, LP HLJ29L2 continuously produced lactic acid, acetic acid, and other organic acids, and the pH dropped rapidly. The pH and TTA did not change much after 12 h. It has been reported that lactic acid bacteria do not continue to produce acid after 12 h.

### 3.6. Growth of LP HLJ29L2 and Yeast during Sourdough Fermentation

During fermentation, the growth and metabolism of LP HLJ29L2 and yeast have an important impact on the sourdough, and this is closely related to changes in the pH and organic acids of the sourdough [30]. The results concerning the growth of LP HLJ29L2 and yeast of the sourdoughs during fermentation are depicted in Figure 3c,d. LP HLJ29L2 exhibited a rapid growth trend between 0 and 10 h. After this, growth was characterized by a stationary phase between 10 and 15 h, then the viable LP HLJ29L2 counts slightly declined. Finally, after 19 h of fermentation, the log of colony forming units per gram (log cfu/g) in DN-1 sourdough was about 8.5. During the early stages, it is possible that nutrients were adequate (sugar and free amino acids), so LP HLJ29L2 could quickly adapt to the environment of sourdough to grow. At the same time, with the decrease in pH and the increase in organic acids, between 0 and10 h of fermentation, LP HLJ29L2 grew fast, which was consistent with the literature [31]. After 10 h, the growth of LP HLJ29L2 was restricted and the number was reduced. On the one hand, it could be that nutrients in sourdough were used by microorganisms and could not be supplemented in time. On the other hand, the pH value in the sourdough was too low, which was not suitable for microorganisms to grow.

The viable yeast count rapidly changed (Figure 3d), and it was similar to the growth of LP HLJ29L2. In yeast sourdough, the rapid growth of yeast was due to sufficient nutrients. After 6 h, the number of yeast dropped rapidly and the curve gradually became flat. The main reason was that the nutrients were consumed in large amounts and could not be supplemented in time. The microorganisms in sourdough, LP HLJ29L2 and yeast, were in a symbiotic relationship [32]. During 0–12 h in DN-1 sourdough, the growth of yeast slowly increased, and the number of yeast dropped rapidly after 12–19 h. This might be due to the decrease in pH.

### 3.7. Organic Acid Measurements in Sourdoughs

The following organic acids, lactic acid, acetic acid, citric acid, fumaric acid, and malic acid, were measured in the sourdough during fermentation (Figure 4). The content of all five organic acids was the highest in the DN-1 sourdough. Among these organic acids, malic acid in all the three sourdoughs was present in the highest content, and lactic acid content continuously increased in the DN-1 sourdough. During the fermentation process, the organic acid of the N-L-Y and yeast sourdough did not change much, and the total acid content remained stable. During the fermentation time of 6–12 h, lactic acid and acetic acid were rapidly produced and accumulated in the DN-1 sourdough, while malic acid gradually decreased. After 12 h, the lactic acid content continued to increase, and the content of other acids also changed but not strongly. Finally, the values of lactic acid, malic acid, and acetic acid were 15.1, 8.6, and 0.8 mg/g, respectively. The results were consistent with the pH and TTA curves of sourdoughs (Figure 3b). Large amounts of acetic acid and lactic acid were responsible for the decrease in the pH of DN-1 sourdough. Citric acid and fumaric acid were present in low concentrations, similar to those in the N-L-Y and yeast sourdough. The concentrations of citric acid and fumaric acid were significantly different among the bacterial strains [33]. The citric acid and fumaric acid contents were higher in DN-1 sourdough than in N-L-Y sourdough.

## 4. Conclusions

LP HLJ29L2 grew fast and produced a large amount of organic acids in the sourdough during co-fermentation with LP HLJ29L2 and yeast. This significantly reduced the pH and activated the protease activity of the sourdough, which promoted the degradation of proteins. Co-fermentation improved the protein digestibility and the content of free amino acids in the crackers made from sourdough. This study showed a significant role of co-fermentation with lactic acid bacteria and yeast in proteolysis during sourdough fermentation as well as protein digestibility in fermented crackers.

## Figures and Tables

**Figure 1 foods-12-00555-f001:**
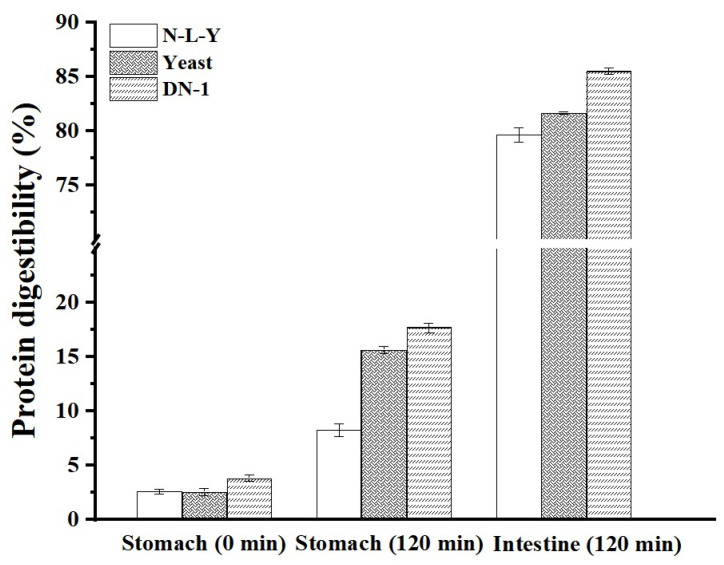
Protein digestibility of crackers in the stomach and intestine. N-L-Y: Crackers made without yeast and LP HLJ29L2 co-fermentation; yeast: Crackers made with yeast fermentation alone; DN-1: Crackers made with LP HLJ29L2 and yeast co-fermentation.

**Figure 2 foods-12-00555-f002:**
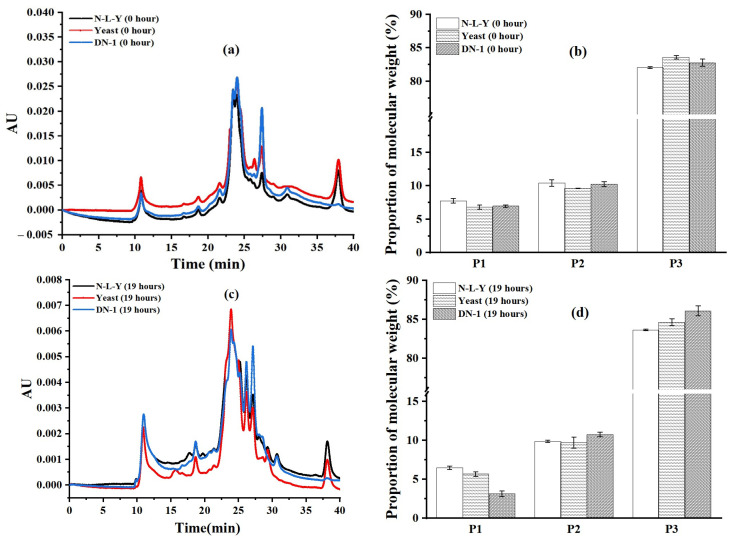
Molecular weight distributions (**a**) and proportion of molecular weights (**b**) of SDS-soluble proteins in the sourdough after fermentation for 0 h. Molecular weight distributions (**c**) and proportion of molecular weights (**d**) of SDS-soluble proteins in the sourdough after fermentation for 19 h. P1 (large glutenin polymers with Mw larger than 91 kDa); P2 (large monomers with Mw from 10 kDa to 91 kDa); and P3 (peptides and amino acids with Mw under 10 kDa). N-L-Y: The dough without yeast and LP HLJ29L2 co-fermentation; yeast: the dough with yeast fermentation alone; DN-1: the sourdough with LP HLJ29L2 and yeast co-fermentation.

**Figure 3 foods-12-00555-f003:**
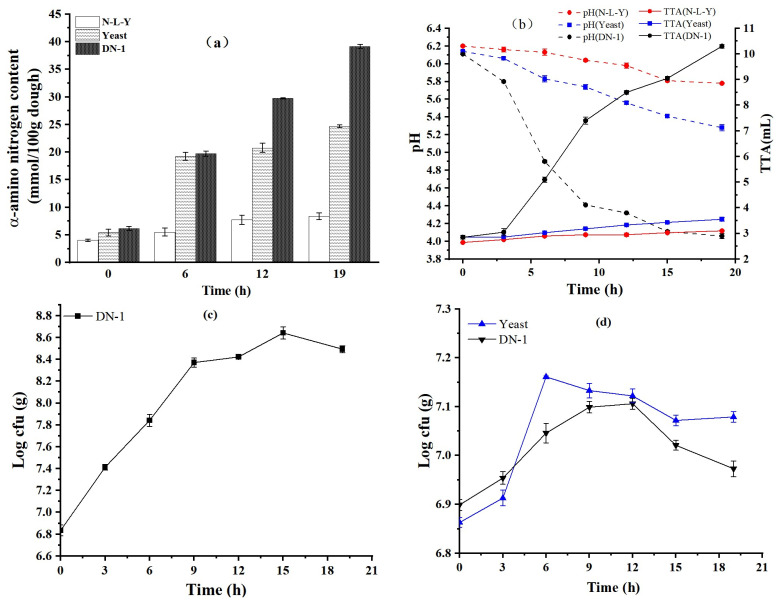
Changes of α-amino nitrogen content (**a**), pH and TTA (**b**), the growth of LAB in DN-1 sourdough (**c**), and the growth of yeast in DN-1 and yeast sourdough (**d**) during fermentation.

**Figure 4 foods-12-00555-f004:**
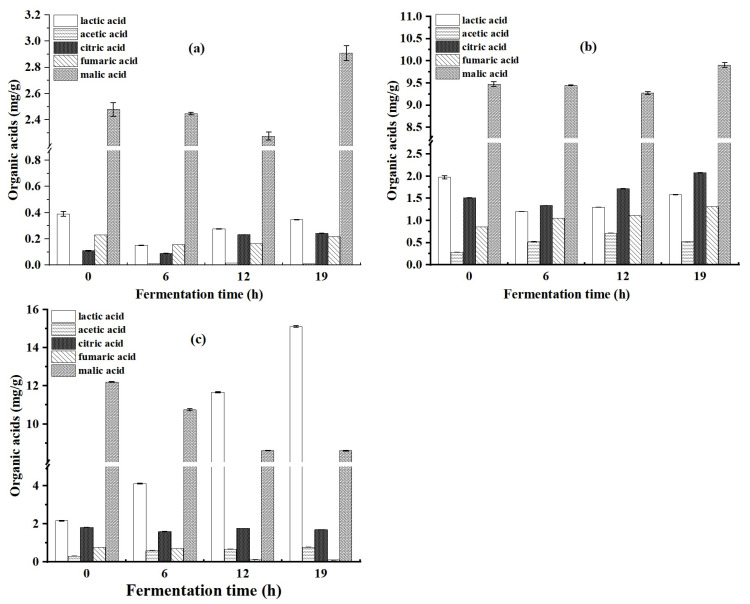
Effect of fermentation on the organic acid content of sourdoughs. (**a**) N-L-Y dough; (**b**) yeast dough; (**c**) DN-1 sourdough.

**Table 1 foods-12-00555-t001:** GMP content and depolymerization degree in different fermented doughs.

Sourdoughs	GMP (g/100 g)		Depolymerization Degree (%)
	0 (Hour)	19 (Hour)	
N-L-Y	0.59 ± 0.02 ^a^	0.55 ± 0.06 ^a^	6.77
Yeast	0.63 ± 0.01 ^b^	0.58 ± 0.02 ^b^	7.94
DN-1	0.60 ± 0.03 ^b^	0.51 ± 0.04 ^a^	15.00

Note: Results are presented as average ± standard deviation (*n* = 3). Different letters within a line indicate significant differences (Tukey’s test, *p* < 0.05). N-L-Y: The dough without yeast and LP HLJ29L2 co-fermentation; yeast: the dough with yeast fermentation alone; DN-1: the sourdough with LP HLJ29L2 and yeast co-fermentation.

**Table 2 foods-12-00555-t002:** Free amino acid content of three kinds of fermented crackers.

Free Amino Acids (mg/g)	N-L-Y	Yeast	DN-1
Asp	10.42 ± 0.04 ^a^	16.75 ± 0.29 ^c^	11.25 ± 0.37 ^b^
Glu	16.95 ± 0.10 ^a^	42.52 ± 0.65 ^c^	33.21 ± 0.43 ^b^
Ser	1.17 ± 0.05 ^a^	1.78 ± 0.13 ^a^	2.19 ± 0.23 ^b^
His	3.99 ± 0.13 ^a^	4.90 ± 0.14 ^b^	7.37 ± 0.31 ^c^
Gly	8.03 ± 0.06 ^b^	4.16 ± 0.23 ^a^	9.41 ± 0.61 ^c^
Thr	5.25 ± 0.03 ^a^	6.05 ± 0.20 ^a^	8.41 ± 0.08 ^b^
Arg	10.39 ± 0.17 ^b^	7.90 ± 0.01 ^a^	20.79 ± 0.60 ^c^
Ala	17.65 ± 0.01 ^a^	16.90 ± 0.50 ^a^	20.37 ± 0.30 ^b^
Tyr	9.13 ± 0.02 ^b^	7.18 ± 0.12 ^a^	8.40 ± 0.27 ^b^
Cys	8.22 ± 0.11 ^b^	0.27 ± 0.09 ^a^	1.04 ± 0.08 ^a^
Val	1.24 ± 0.07 ^a^	10.41 ± 0.40 ^b^	13.23 ± 0.18 ^c^
Met	10.31 ± 0.03 ^c^	2.31 ± 0.18 ^a^	3.66 ± 0.33 ^b^
Phe	2.16 ± 0.01 ^a^	4.75 ± 0.03 ^b^	8.12 ± 0.20 ^c^
Ile	7.11 ± 0.01 ^c^	3.40 ± 0.09 ^a^	4.97 ± 0.54 ^b^
Leu	5.34 ± 0.11 ^b^	4.53 ± 0.36 ^a^	11.53 ± 0.70 ^c^
Lys	10.73 ± 0.16 ^c^	4.68 ± 0.30 ^a^	8.39 ± 0.95 ^b^
Pro	13.48 ± 0.09 ^a^	15.09 ± 0.71 ^b^	22.48 ± 1.39 ^c^
Gaba	5.68 ± 0.01 ^a^	8.81 ± 0.11 ^b^	13.68 ± 0.46 ^c^
Total amino acids	147.26 ± 0.10 ^a^	162.40 ± 0.09 ^b^	208.51 ± 0.26 ^c^
Essential amino acids	51.28 ± 0.08 ^b^	43.31 ± 0.07 ^a^	66.71 ± 0.14 ^c^
E/T (%)	32.61	26.67	31.99

Notes: Data are average ± error of triplicates. Different letters in the same column indicate significant differences when Tukey’s test was applied (*p* < 0.05). N-L-Y: The cracker without yeast and LP HLJ29L2 co-fermentation; yeast: the cracker with yeast fermentation alone; DN-1: the cracker with LP HLJ29L2 and yeast co-fermentation.

## Data Availability

The data presented in this study are available on request from the corresponding author. The data are not publicly available due to plans for use in other future works.

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
