# Peer review of "The Effect of Co-Fermentation with Lactobacillus plantarum HLJ29L2 and Yeast on Wheat Protein Characteristics in Sourdough and Crackers"

_foods, 2023, doi:10.3390/foods12030555_

Round 1
Reviewer 1 Report
This work investigates the influence of fermentation of cracker dough with yeasts and co-fermentation of yeast and L. plantarum with emphasis on protein degradation in an acidic medium. The influence of an acidic medium on protein degradation is well known. This influence of co-fermentation of yeast and L. plantarum is also not seen, as there is no dough that was fermented only with L. plantarum. Perhaps the results would be the same without yeast if fermented with bacteria only?
I suggest changing the name of the samples - I would not call a sample without yeast and L. plantarum bacteria sourdough, but maybe just dough? A yeast-only sample would not be "yeast sourdough" - sourdough is a name that implies lactic acid bacteria, perhaps yeast dough? In the case of a cracker, that's fine (e.g., L 168)
The counting of CFU during sourdough fermentation is not properly described. How to determine yeasts on plates with cycloheximide (L 154)?
L158 - SPSS ???
All designations of figures and tables should be more detailed, abbreviations must be described in the designation of figures and tables independently of the main text.
In Figure 3, the abbreviation appears DY -1??? I do not even understand part c and d of Figure 3 (L 269).
The Appendix (L 78) is not available for me to view, so the entire description of the preparation of yeasts and bacteria is missing - are these laboratory or commercial cultures?
It is necessary to revise the entire text additionally.
Reviewer 2 Report
The present paper analyses the effect of co-fermentation with Lactobacillus plantarum HLJ29L2 and yeast on wheat protein characteristics in sourdough and crackers. The paper has aspects of novelty and may be of interest to the scientific community.
The title of the manuscript should be improved (The effect of co-fermentation with Lactobacillus plantarum HLJ29L2 and yeast on wheat protein characteristics in sourdough and crackers)
The novelty of the research must be emphasized in the abstract as well as in the conclusions.
Introduction. The introduction should focus more on the problem which is solved in the paper and emphasizes what constitutes research novelty, based on more recent publications.
Methods
line 78. The first sentence is not clear (what is presented in an appendix?)
line 107. the formula must be of better quality (not uploaded as a picture, but written)
line 110. Free amino acid analysis
line 117. How was prepared sourdough powder?
line 121. Specify filter type
line 122. use " the precipitate" instead of 'the precipitation'. specify procedure
line 136. What means "A calibration curve was prepared for each determination"?
line 138. TTA - total titratable acidity
lines 141-142. "The TTA represents the volume of 0.1 mol/L NaOH consumed" for what sample amount? (ml/100 g or ml/10 g?)
line 146. specify filter type
line 153. Why such a temperature (37C) was selected for incubation?
Results
lines 203-204. Explain which amino acids are required for the growth of LAB and yeast.
lines 287-288. Expand the explanation for the last sentence.
References. Improve the numbering of references.
Conclusions. The conclusions you include in the paper should be more succinct and precise. Rather than repeating parts of the paper, use this last paragraph to wrap up what you want readers to remember most.
Figures. Increase the resolution, where is possible, and the font size for all figures.
